# Lipid Polymorphism of the Subchloroplast—Granum and Stroma Thylakoid Membrane—Particles. I. ^31^P-NMR Spectroscopy

**DOI:** 10.3390/cells10092354

**Published:** 2021-09-08

**Authors:** Ondřej Dlouhý, Uroš Javornik, Ottó Zsiros, Primož Šket, Václav Karlický, Vladimír Špunda, Janez Plavec, Győző Garab

**Affiliations:** 1Group of Biophysics, Department of Physics, Faculty of Science, University of Ostrava, 710 00 Ostrava, Czech Republic; ondrej.dlouhy@osu.cz (O.D.); vaclav.karlicky@osu.cz (V.K.); vladimir.spunda@osu.cz (V.Š.); 2Slovenian NMR Center, National Institute of Chemistry, SI-1000 Ljubljana, Slovenia; uros.javornik@ki.si (U.J.); primoz.sket@ki.si (P.Š.); janez.plavec@ki.si (J.P.); 3Photosynthetic Membranes Group, Plant Light Perception and Utilization Research Unit, Institute of Plant Biology, Biological Research Centre, Eötvös Loránd Research Network, 6726 Szeged, Hungary; zsiros.otto@brc.hu; 4EN-FIST Center of Excellence, SI-1000 Ljubljana, Slovenia; 5Laboratory of Ecological Plant Physiology, Domain of Environmental Effects on Terrestrial Ecosystems, Global Change Research Institute of the Czech Academy of Sciences, 603 00 Brno, Czech Republic; 6Faculty of Chemistry and Chemical Technology, University of Ljubljana, SI-1000 Ljubljana, Slovenia

**Keywords:** ^31^P-NMR, bilayer membrane, DEM—dynamic exchange model, grana, H_II_ phase, isotropic phase, non-bilayer lipids, non-lamellar lipid phases, thylakoid membranes, structural flexibility

## Abstract

Build-up of the energized state of thylakoid membranes and the synthesis of ATP are warranted by organizing their bulk lipids into a bilayer. However, the major lipid species of these membranes, monogalactosyldiacylglycerol, is a non-bilayer lipid. It has also been documented that fully functional thylakoid membranes, in addition to the bilayer, contain an inverted hexagonal (H_II_) phase and two isotropic phases. To shed light on the origin of these non-lamellar phases, we performed ^31^P-NMR spectroscopy experiments on sub-chloroplast particles of spinach: stacked, granum and unstacked, stroma thylakoid membranes. These membranes exhibited similar lipid polymorphism as the whole thylakoids. Saturation transfer experiments, applying saturating pulses at characteristic frequencies at 5 °C, provided evidence for distinct lipid phases—with component spectra very similar to those derived from mathematical deconvolution of the ^31^P-NMR spectra. Wheat-germ lipase treatment of samples selectively eliminated the phases exhibiting sharp isotropic peaks, suggesting easier accessibility of these lipids compared to the bilayer and the H_II_ phases. Gradually increasing lipid exchanges were observed between the bilayer and the two isotropic phases upon gradually elevating the temperature from 5 to 35 °C, suggesting close connections between these lipid phases. Data concerning the identity and structural and functional roles of different lipid phases will be presented in the accompanying paper.

## 1. Introduction

In plants, the light reactions of photosynthesis occur in chloroplast thylakoid membranes (TMs), flattened lipid vesicles which separate the inner luminal and the outer, stroma-side aqueous phases. These membranes embed the two photosystems (PSs) PSII and PSI, containing the photochemical reaction center core complexes and their light-harvesting antenna proteins (LHCII and LHCI, respectively), the cytochrome b_6_f complex and some additional components of the electron transport system, and the ATP-synthase. The operation of the linear electron transport system leads to the evolution of molecular oxygen, released to the atmosphere, and the synthesis of NADPH, providing reducing equivalents for CO_2_ fixation. The primary charge separation in the photo-chemical reaction centers and the consecutive vectorial charge transport generate the energized state of the TM, an electrochemical potential gradient (Δμ_H_^+^), or proton-motive force, which is utilized for the synthesis of ATP [1].

The build-up of Δμ_H_^+^ and the production of ATP are warranted by the organization of the TM lipids into a bilayer, which is impermeable to water, most water-soluble molecules, and ions [2,3,4]. However, in TMs the major lipid species, the monogalactosyldiacylglycerol (MGDG) (constituting about half of their lipid content [5]) is a non-bilayer lipid. Only about the other half of TM lipids—digalactosyldiacylglycerol (DGDG, ∼25–30%), sulfoquinovosyldiacylglycerol (SQDG, ∼10–15%) and phosphatidylglycerol (PG, ∼10–15%)—are capable of forming bilayers in aqueous media [4]. In contrast, MGDG—similar to other non-bilayer lipid species in different biological membranes [6,7,8]—prefers to adopt non-lamellar or non-bilayer phases, such as the inverted hexagonal (H_II_), isotropic, and cubic phases [4,9]. This holds true for total lipid mixtures of TMs, which are unable to maintain a stable bilayer phase under physiologically relevant conditions [4]; instead, they form stalks, non-bilayer molecular assemblies [10]. Indeed, as shown via small-angle neutron scattering (SANS), TM lipid mixtures readily undergo phase transitions, governed mainly by their hydration state [11]. On the other hand, the in vitro association of MGDG and LHCII can lead to the formation of large, ordered lamellar structures [12]—demonstrating that non-bilayer lipids can be forced to adopt a bilayer organization [13]. It has also been shown that the addition of MGDG to loosely stacked lipid-LHCII membranes substantially increase their structural flexibility, and, particularly, their ability to undergo light-induced reversible reorganizations [14]—demonstrating that lipids with a high non-bilayer propensity lend the membranes additional structural plasticity.

Different membrane models have been proposed, which—in contrast to the ‘standard’, fluid mosaic membrane model [2,15]—take into consideration the presence of non-bilayer lipids in biological membranes. According to the lateral pressure model (LPM), non-bilayer lipids, due to their conical shape, increase the lateral pressure in the acyl chain region of the bilayer and decrease the pressure in the lipid headgroups region [16]. By this means, non-bilayer lipids can modulate protein functions [17], a mechanism recently proposed to be involved in the regulation of the light-harvesting vs. energy-dissipation functions of LHCII [18]. In the flexible surface model (FSM), membrane–protein conformations are proposed to be governed by the curvature elastic energy profiles, determined largely by the balance of curvature and hydrophobic forces in lipid−protein interactions [19]. In general, lipid mixtures with strong non-bilayer propensity might be found at the boundary of lamellar-to-non-bilayer phase transition. Hence, the FSM, similarly to the LPM, considers the bilayer membrane being in frustrated state—but both the LPM and the FSM postulate that non-bilayer phases are present only locally and transiently in the bilayer. Nevertheless, these frustrated states might play significant physiological roles, since “the bilayer must not be too stable because that would tend to limit protein dynamics” [20].

The dynamic exchange model (DEM) proposes to extend the structural flexibility of membranes in the transmembrane direction by assuming the coexistence and close association of the bilayer membrane with non-bilayer lipid phases and a dynamic equilibrium between these different phases [21,22]. This model is based on the Janus face of lipids possessing non-bilayer propensity. On the one hand, they can be incorporated in the bilayer with the help of proteins ([18], see above). On the other hand, they readily segregate from it: large protein-free areas of membranes are expected to form transient structures which are then expelled from the membranes [3,21]. Given the limited volume of the aqueous-phase compartments, especially on the luminal side of TMs, it is a close assumption that the bilayer phase is in dynamic equilibrium with its closely associated non-bilayer phase(s). Recently, the role of plastoglobuli has been hypothesized to participate in a dynamic lipid exchange with the outer leaflet of the bilayer of the TM [23].

The coexistence of the bilayer phase with an isotropic phase in fully functional isolated plant thylakoid membranes has been first demonstrated by Krumova and coworkers [24] by employing ^31^P-NMR spectroscopy. The use of this technique for fingerprinting the phase behavior of phospholipids in native and artificial membranes has been thoroughly documented [25,26,27]. In TMs, PG is the only phospholipid, about 60% of which is present in the bulk phase [28]; it has been shown to be a sensitive marker of the lipid phases [29,30]. It is important to stress that there is no lateral heterogeneity of the distribution of PG and all other lipids in the bulk phase of thylakoid membranes [31]. Further, data from molecular dynamics simulations on thylakoid lipids also revealed “a well-mixed system in both the lamellar and inverted hexagonal state” [10].

Our more recent ^31^P-NMR spectroscopy experiments have revealed the presence of two isotropic phases and an H_II_ phase, in addition to the bilayer phase [32]. The heterogeneity of the packing of lipids has also been confirmed using time-resolved fluorescence spectroscopy using the lipophylic dye Merocyanine 540 [32,33]. The lipid phases of TMs have been shown to undergo different, largely reversible reorganizations, induced by varying the temperature, the pH, and the ionic and osmotic strengths of the medium. These variations in the polymorphic phase behavior of lipids were associated with characteristic changes in the macro-organization of proteins, were correlated with the fine-tuning of the permeability of membranes, and appeared to regulate the photoprotective activity of the water-soluble lipocalin-like enzyme, the violaxanthin de-epoxidase (VDE) [34,35,36]. These observations are in harmony with the basic predictions of the DEM of TMs and with the tentative assignments of the non-lamellar lipid phases. The H_II_ phase was proposed to be formed by lipids expelled from the bilayer, the two isotropic phases were hypothesized to indicate the presence of VDE-lipid (and possibly other lipocalin-lipid) assemblies, and to arise from the interwoven network of granum and stroma TMs, containing regions where membranes fuse together [37] or divide by forking [38,39]. However, the origin of the different lipid phases, in terms of structural entities of the TM system, has remained elusive.

Here, using ^31^P-NMR spectroscopy, we investigated the lipid polymorphism of the two main membrane constituents of plant TM systems, the granum and stroma TMs. These sub-chloroplast membrane particles possess strikingly different protein compositions [40], which, as inferred from the LPM and the FSM, might require different lipid polymorphisms for their optimal functioning. However, we show that both of these membrane fractions exhibit four distinct lipid phases, resembling the polymorphism of TMs; lipase and heat treatments of granum and stroma TMs induced marked and specific effects on the lipid phases. In the accompanying paper, to obtain information on the origin and significance of TM lipid polymorphism, we examined the effects of the same treatments on the molecular organization and functional activity of the photosynthetic machineries and analyzed the ultrastructural features of the granum and stroma sub-chloroplast membrane particles.

## 2. Materials and Methods

### 2.1. Isolation of Granum and Stroma Thylakoid Membranes

The granum and stroma sub-chloroplast thylakoid membrane particles were isolated by digitonin fragmentation of spinach TMs, followed by differential centrifugation, using modified protocols of [41,42], respectively. Spinach leaves, purchased from the local market, were washed in chilled deionized water and kept for overnight at 4 °C in darkness before use. About 250 g leaves were homogenized in 200 mL medium containing 50 mM Tricine-KOH buffer (pH 7.8), 300 mM sorbitol, 25 mM NaCl, 25 mM KCl, and 5 mM MgCl_2_. The crude extract was filtered through four layers of perlon net, followed by a 1.5 min centrifugation at 3000× *g*. The pellet was suspended for 1.5 min in 30 mL of 10 mM MgCl_2_, after which 30 mL 40 mM Tes-KOH buffer (pH 7.8), containing 500 mM sorbitol, 50 mM KCl, and 50 mM NaCl, was added. After centrifugation at 3500× *g* for 10 min, the pellet was resuspended in 20–30 mL of 20 mM Tes-KOH buffer (pH 7.8), containing 250 mM sorbitol, 25 mM KCl, 25 mM NaCl, and 5 mM MgCl_2_. Digitonin (Merck, Darmstadt, Germany; twice recrystallized from ethanol) was applied in this medium at a final concentration of 0.2% (*w*/*v*) and a digitonin/Chl (*w*/*w*) ratio of 1 (Chl, chlorophyll). The mixture was stirred for 30 min in the dark at 4 °C, followed by a 3-fold dilution with the same medium. After centrifugations for 30 min at 5000× *g*, the supernatant was further centrifuged for 30 min at 10,000× *g*. The pellet (D10, granum TMs) was resuspended in a medium containing 5 mM Tes-KOH (pH 7.8), 20 mM Tricine-KOH (pH 7.8), 20 mM NaCl, 20 mM KCl, and 5 mM MgCl_2_. The remaining supernatant was further centrifuged for 30 min at 50,000× *g* and the pellet was resuspended in the same medium. The final centrifugation of the supernatant was carried out for 60 min at 130,000× *g* to obtain D130, the stroma TMs, which were resuspended in small volumes of the same medium as used for D10. The Chl content of the isolated particles, determined according to [43], were always higher than 5 mg Chl (a + b) mL^−1^. The Chl a/b ratios of D10 and D130 were 3.05 ± 0.23 (*n* = 4) and 9.68 ± 2.7 (*n* = 4), respectively. All isolation procedures were performed at 4 °C in dim green laboratory light and the samples were stored at −80 °C until use.

### 2.2. Lipase Treatments

In the experiments on lipase-treated granum and stroma TMs, a substrate non-specific [44], general tri-, di-, and monoglyceride hydrolase—lipase from wheat germ (L3001, Sigma-Aldrich, Burlington, MA, USA) was used. A stock solution of 0.5 U µL^−1^ activity was prepared in MilliQ water, and volumes containing the desired activity of the lipase were added into the sample and thoroughly mixed prior to the start of the measurement.

### 2.3. ^31^P-NMR Measurements

^31^P-NMR measurements were performed as described earlier [35]. Spectra were recorded using Avance Neo 600 MHz NMR spectrometer (Bruker, Billerica, MA, USA) equipped with a BBFO probe tuned at the resonance frequency of the ^31^P nucleus. Circa 1 mL of sample was placed into a 5 mm outer diameter NMR tube. Our earlier experiments have shown that—due to the very high density of the TMs—no magnetic orientation of the sample occurs [24]. (N.B.: intact TMs possess considerably higher diamagnetic anisotropy values than the sub-chloroplast particles, and thus TMs are easier to align in an external magnetic field than their fragments.) Spectra were recorded using a 40° rf pulse, an inter-pulse time of 0.5 s and no ^1^H-decoupling. The temperature was controlled to within ±0.1 °C. Chemical shifts were referenced externally to 85% solution of H_3_PO_4_ in water.

For saturation experiments, a low power RF pulse was applied at the selected chemical shift for 0.3 s, followed by a 40° pulse and 0.2 s of acquisition, for a repetition time of 0.5 s. The power of the pre-saturation pulse was adjusted according to the peaks of interest—the RF pulse field strength was 80 Hz for pre-saturation of lamellar and H_II_ phase and 40 Hz for the isotropic phases.

During averaging of the ^31^P-NMR spectra, weighting factors were applied to correct for the different chlorophyll contents of the samples, and were normalized to 10 mg Chl (a + b) mL^−1^ in all cases. Weighting factors were also applied if the number of scans of the spectra used for averaging were not the same.

The ^31^P-NMR spectra were processed using TopSpin (Bruker, Billerica, MA, USA) software, mathematical deconvolution was carried out using DMfit software (Dominique Massiot, Orléans, France) [45]—using the spectral distributions for lamellar, H_II_ and isotropic phases [25,27]. The figures were plotted using MatLAB R2018a (the MathWorks, Inc., Portola Valley, CA, USA) with a Spectr-O-Matic toolbox (Dr. Petar H. Lambrev, Szeged, Hungary) for analysis of spectroscopy data.

## 3. Results

### 3.1. ^31^P-NMR Fingerprints of Lipid Phases in Isolated Granum and Stroma Thylakoid Membranes

The ^31^P-NMR spectra of isolated granum and stroma thylakoid membranes recorded at 5 °C revealed that the signals originated from several different chemical environments of the phosphorous nucleus (Figure 1). As already pointed out in the case of isolated intact thylakoid membranes [24], the signals arise predominantly from PG, the only phospholipid molecules of TMs. Although the spectra of the two sub-chloroplast TM particles were not identical, and in both cases the spectral distributions varied from batch to batch, the basic features remained very similar to one another and resembled the spectra of isolated intact TMs [35,36].

As shown in Figure 1, the spectra could be deconvoluted to four component spectra. Similar to intact TMs, both membrane particles exhibited a well-defined lamellar phase, peaking around −10 ppm and displaying an asymmetric shape extending to the low-field side. A spectral shape with reversed symmetry, characteristic of the H_II_ phase, was also observed with a peak position at around 20 ppm. This peak appeared to be somewhat shifted compared to TMs, where it was typically found at ~25–30 ppm [36]. The spectra also contained two sharp, symmetric bands (I_1_ and I_2_) in the region between about 2.5 and 4.5 ppm; these resonances are of isotropic origin. The relative contributions of the four different phases were also very similar in the granum and stroma TMs (Figure 1, Panels b and d). The lamellar phases displayed large integrated areas; somewhat surprisingly, equally large or somewhat (albeit statistically insignificantly) larger average integral values were obtained for the H_II_ phases. Although the I_2_ peak was usually weaker than I_1_, and in some cases could only be discerned as a shoulder (as in Figure 1a), it was clearly present in all samples. The peak positions of I_1_ and I_2_ were essentially the same in the granum and stroma TMs, 2.66 ± 0.07 and 2.68 ± 0.16, and 3.98 ± 0.40 and 4.19 ± 0.13, respectively; and only the half-bandwidths appeared to be somewhat narrower in the stroma TMs (Table 1).

### 3.2. Saturation-Transfer Experiments

To probe the spectral shapes of different phases, and to test the validity of the above mathematical deconvolution of the ^31^P-NMR spectra, we performed saturation-transfer experiments, i.e., applied saturating pulses, to suppress contributions from different phases, at selected frequencies at or near the peak positions of the spectral components tested [24,46]. This approach was practicable due to the relatively high stability of the sub-chloroplast particles, as shown by the data in Appendix A. It can be seen that the spectra recorded in the first and second 30 min of the acquisition time were essentially identical; in addition, the lamellar phases were retained with very little decrease during several hours, in contrast to intact TMs under similar conditions [34]; further, the isotropic phases increased only moderately during the time of up to several-hours long measuring periods.

As shown in Figure 2a, a saturating pulse applied at −12 ppm eliminated the lamellar-phase signal of granum TMs, while it exerted virtually no effect on the isotropic region. At the same time, in harmony with the sizeable overlap of the H_II_-phase spectral component with the lamellar phase, the intense pulse at −12 ppm also caused a moderate decrease in the H_II_ signal. Vice versa, when the (nearly) saturating pulse was applied at 20 ppm (i.e., at the peak position of the H_II_-phase), a decrease in the intensity of the lamellar-phase component was observed (Figure 2b). With the same pulses, the isotropic phases did not decrease; in fact, here, the I_1_ intensity increased, which evidently could be accounted for by the time lapse between recording the control and the saturation-transfer spectra. The two isotropic peaks, I_1_ and I_2_ could also be suppressed by selective pulses at characteristic frequencies, near the peak positions (Figure 2 Panels c and d). The selectivity of suppressions was high, if taking into account the small frequency difference between the two peaks. Saturation-transfer experiments on stroma TMs yielded very similar results as obtained for granum TMs. In particular, the ^31^P-NMR signals of the lamellar and H_II_ phases as well as those associated with the intense I_1_ phase and the somewhat weaker I_2_ phase could be suppressed with reasonable degree of selectivity (Figure 3 Panels a–d).

In general, these data, in line with the above deconvolution analyses, show that the sub-chloroplast TMs at 5 °C contain four lipid phases, each of which originate from distinct structural entities. Further, it can also be seen that the spectral shapes of the different phases obtained by using the mathematical deconvolution, and those which can be inferred from the saturation transfer experiments are in good agreement with one another.

### 3.3. Temperature Dependences

The lamellar phase of granum TMs was gradually destabilized upon increasing the temperature stepwise from 5 to 15; from 15 to 25 and then to 35 °C; shifts and broadenings were also observed in the isotropic region (Figure 4). Deconvolution of the spectra (Panels b–e of Figure 4), and an inspection of the temperature-dependent variations of the integrated areas of the component spectra (Appendix A) revealed a substantial increase in the intensity of the I_1_ phase at the expense of the lamellar and H_II_ phases. At the same time, the sharp peak of I_1_, at ~2.6 ppm gradually shifted to about 1 ppm and broadened considerably. For I_2_, no such shift was observed, and the broadening appeared only at 35 °C (Figure 4e). Very similar trends were observed in stroma TMs (Figure 5a): a gradual loss of the lamellar phase (Figure 5b); some diminishment of the H_II_ phase—albeit its overall low intensity in the given samples makes this diminishment questionable (Figure 5c); and more prominent changes in I_1_ (Figure 5d) compared to I_2_ (Figure 5e).

In general, these data show that the different lipid phases do not react uniformly to the increase of the temperature; most remarkably, elevating the temperature destabilized the lamellar phases both in the granum and the stroma TMs; different responses of I_1_ and I_2_ should also be noticed.

### 3.4. Effect of Wheat Germ Lipase

To test the potentially distinct lipase sensitivity of the different lipid phases of granum and stroma TMs, we performed ^31^P-NMR spectroscopy measurements in the absence and presence of different concentrations of wheat-germ lipase. As shown in Figure 6a, whereas the H_II_ phase of granum TMs was essentially insensitive to the lipase and the lamellar phase was only marginally affected by 5 U lipase treatment, the sharp I_1_ isotropic phase was essentially eliminated and replaced by a broad band—indicating the appearance of a large, immobilized molecular assembly replacing a highly mobile lipid-containing domain. Again, as for the thermal treatment, the I_2_ phase was less sensitive to the lipase treatment. These effects depended on the concentration of the lipase and progressed with increasing the enzyme concentration—as also reflected in the deconvoluted component spectra (Figure 6b–e). Because the broad isotropic band largely overlapped the bands of the lamellar and H_II_ phases, we applied saturation pulses, and confirmed the existence of these latter phases even after the lipase treatments (Appendix A). These experiments also confirmed the appearance of the broad isotropic phase, which could be suppressed, with reasonable selectivity, with a pulse at 2.8 ppm. It is clear that already after treating the membranes with the lipase at 5 U, the isotropic signals were replaced with a broad featureless peak, which was broadened into the baseline upon further addition of the lipase. The observed broadening of the signal is indicative of a reduction of T2* relaxation time, which we interpret as the result of a reduction of molecular mobility due to the redistribution of lipids from the isotropic phase to a larger phospholipid formation [47]. Very similar data were obtained on stroma TMs, which appeared to be more susceptible to the same lipase treatment: 2.5 U exerted similar effect on these membranes as 5 U on the grana (Figure 7a). Here, too, the lamellar phase (Figure 7b) and the H_II_ phase (Figure 7c) were only marginally sensitive. In contrast, the two sharp isotropic peaks were essentially eliminated (Figure 7d,e) and the broad resonance attributed to the emergence of a larger formation composed of released or partially cleaved lipid molecules resulting from the action of the lipase was observed.

These data, in good agreement with the data from the saturation transfer experiments (Section 3.2), indicate that the lamellar, H_II_, the two isotropic phases originate from distinct structural entities both in the granum and the stroma TMs, and that the I_1_ isotropic phases in the two membrane particles share very similar susceptibilities to wheat-germ lipase.

## 4. Discussion

In this paper, we investigated the origin of different non-bilayer lipid phases in the context of the lateral heterogeneity of TMs. In chloroplasts of higher plants, the photosynthetic membranes are differentiated into granum and stroma TMs: the cylindrical stacks of grana are interconnected by unstacked stroma membranes, which are helically wound around the granum [38]. Nevertheless, the two types of membranes form one single continuum membrane, enclosing a contiguous inner aqueous (luminal) phase [48]. Whereas the TM lipids are evenly distributed within the plane of the membrane [31], the granum and stroma TMs display strikingly different protein compositions. Indeed, the differentiation of TMs into granum and stroma membranes appears to be governed by sorting of their proteins, via the self-assembly of LHCII-PSII macrodomains, stabilized by stacking [49]. As a consequence, grana are enriched in LHCII-PSII supercomplexes; in contrast, LHCI-PSI and the ATP synthase are found predominantly in the stroma TM and the end-membranes of grana [40].

The marked protein-composition differences between the two types of TMs could, in principle, exert effects on the lipid phase behavior of membranes, especially when considering the predictions of the LPM [16,17] and the FSM [19] on the protein functions. In other terms, the functional differences in the two regions could, in principle, be associated with differences in the lipid phase behavior. However, no such difference was observed (Figure 1). This can be explained by the fact that both the granum and the stroma TMs contain large, compact supercomplexes, robustly organized LHCII-PSII and LHCI-PSI, respectively. These supercomplexes are composed of dozens of tightly packed protein sub-units and highly organized pigment systems, containing more than 200 pigment molecules (chlorophylls and carotenoids) [50,51]. They also bind tens of different lipid molecules—with no apparent distinction between bilayer and non-bilayer lipids, i.e., between PG, SQDG and DGDG, and MGDG, respectively. These non-annular or structural lipids appear to mediate interactions between protein subunits and might fill the grooves in the membrane-embedded protein structures. The structural lipids, and the slowly exchanging annular or shell (boundary) lipids around the supercomplexes do not contribute to the ^31^P-NMR signal of TMs (cf. [22] and references therein).

Regarding the role of the non-bilayer lipid MGDG in the bulk, its presence lends non-bilayer propensity to the lipid mixture, which, in turn, keeps the TMs in a frustrated state—as proposed in all models taking into account the presence of non-bilayer lipids in bilayer membranes: the LPM [16], the FSM [19], and the DEM [22]. This, in general, might be of high physiological significance but, per se, does not explain the observed lipid polymorphism of intact TMs and their constituent fractions. The coexistence of bilayer and non-bilayer lipid phases is one of the basic postulations of the DEM. Thus, our data and our earlier observations concerning the polymorphism of lipid phases in functional thylakoid membranes [24,32] are consistent with this latter model. In this context, it is also worth mentioning that the coexistence of bilayer and non-bilayer lipid phases has also been documented in the other main energy-converting membranes, in functionally active inner mitochondrial membranes [52].

The mathematical deconvolution of the spectra of granum and the stroma TMs (Figure 1), similarly to intact TMs [36], confirmed the presence of two isotropic phases and an H_II_ phase, in addition to the bilayer. A more direct, experimental support on the validity of this spectral deconvolution was obtained from saturation transfer experiments. In particular, by applying saturating pulses at different, characteristic frequencies, we have shown that the signals of the different distinct lipid phases, exhibiting characteristic spectral distributions, can be selectively suppressed by irradiating the samples at or near their peak frequencies (Figure 2 and Figure 3). At the same time, at higher temperatures, we also observed exchanges between different lipid phases (Figure 4 and Figure 5)—in agreement with our earlier works [33,36] and in harmony with the DEM of TMs.

As to the origin of these distinct, yet apparently interconnectable lipid phases, we tested their accessibilities to wheat-germ lipase (Figure 6 and Figure 7). These experiments, while provided no clue concerning the structural entities associated with the non-bilayer lipid phases, have clearly shown that the isotropic phases originate from structural units which are more easily digestible with this lipase than the lipids of the bilayer membrane or of the H_II_ phase. Wheat-germ lipase preferentially destroyed the I_1_ phase and transformed it to a largely immobile molecular assembly.

The elucidation of the questions concerning the identity and physiological roles of the observed non-bilayer phases requires further structural and functional data. This type of experiments, combined with measurements testing the effects of temperature and lipase treatments on the molecular organization and functioning of the photosynthetic machineries in the granum and stroma TMs, have been carried out—and will be reported in the accompanying paper.

## 5. Conclusions

In this work we have shown that the granum and stroma TMs exhibit marked lipid-phase polymorphisms, which are very similar to each other and to the intact TMs: in addition to the bilayer, they exhibit an H_II_ phase and two isotropic phases. These data clearly show that the protein composition of the membranes does not exert a significant effect on the phase behavior of TM lipids. Saturation transfer experiments and lipase treatments at 5 °C have revealed that the different lipid phases are associated with distinct structural entities, which are nevertheless capable of exchanging lipids with one another at higher temperatures. In general, these data are in harmony with the DEM of TMs. In the accompanying paper, by investigating the ultrastructure, spectroscopy, and functional parameters of these preparations, and analyzing the effects of different treatments, we provide insight into the structural identity and origin of different lipid phases and their role in the self-assembly of TM systems; we will also discuss the possible physiological significance of non-bilayer lipid phases in thylakoid membranes and in energy-converting membranes in general.

## Figures and Tables

**Figure 1 cells-10-02354-f001:**
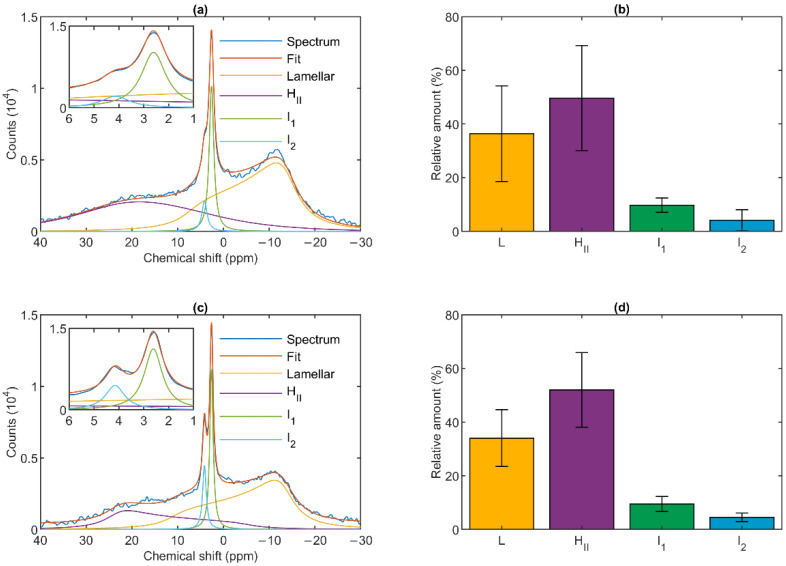
^31^P-NMR spectra (**a**,**c**) and relative intensities (**b**,**d**) of isolated spinach granum (**a**,**b**) and stroma (**c**,**d**) thylakoid membranes at 5 °C. Average of (**a**) five spectra from three batches and (**c**) six spectra from five batches. Integrated areas of the component spectra (**b**,**d**) associated with the different lipid phases, relative to the overall integrated area; mean values ± SD.

**Figure 2 cells-10-02354-f002:**
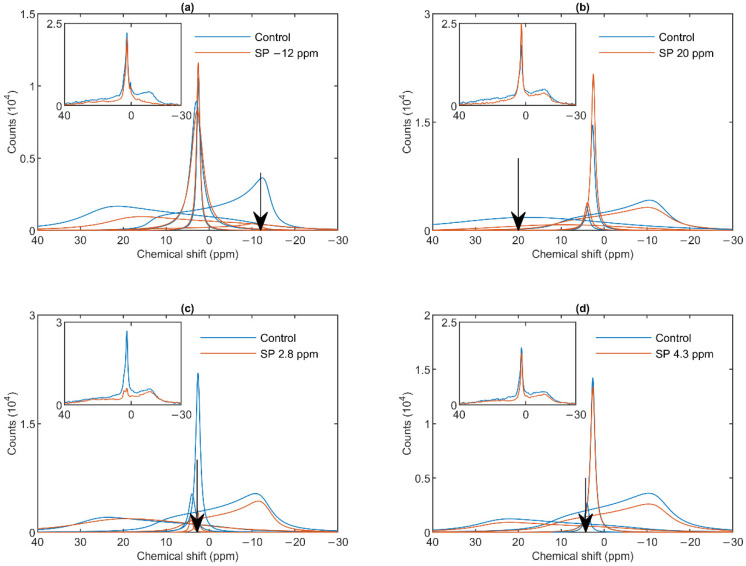
^31^P-NMR spectra of granum thylakoid membranes in the absence (Control, blue curves) and presence (SP, red curves) of saturation pulses applied at different frequencies, as indicated by the arrows, at or close to the peak position of different phases: L, −12 ppm (**a**); H_II_, 20 ppm (**b**); I_1_, 2.8 ppm (**c**); I_2_, 4.3 ppm (**d**). Each panel shows the deconvoluted component spectra and, in inset, the measured spectra. The measurements were performed on different batches; the spectra represent averages from two independent batches with similar polymorphisms; temperature, 5 °C.

**Figure 3 cells-10-02354-f003:**
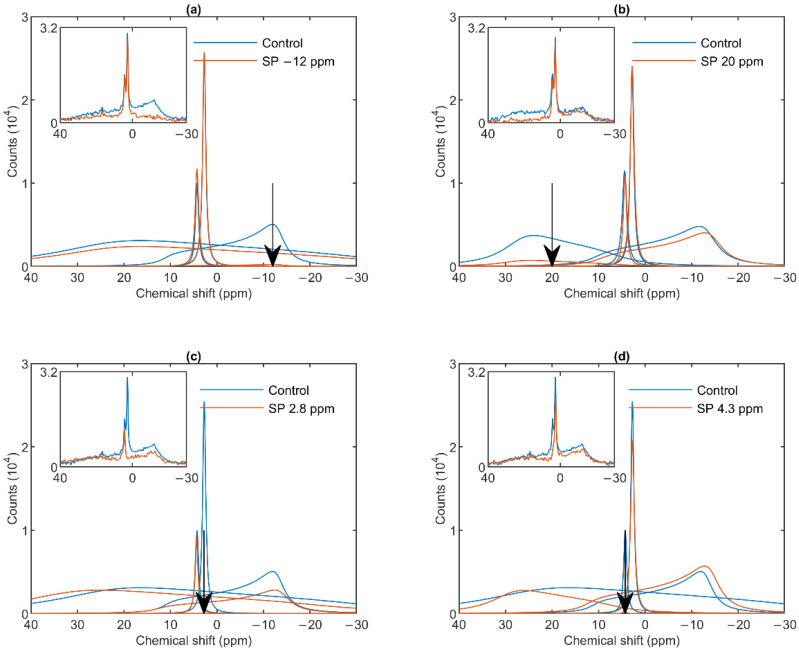
^31^P-NMR spectra of stroma thylakoid membranes in the absence (Control, blue curves) and presence (SP, red curves) of saturation pulses applied at different frequencies, as indicated by the arrows, at or close to the peak position of different phases: L, −12 ppm (**a**); H_II_, 20 ppm (**b**); I_1_, 2.8 ppm (**c**) and I_2_, 4.3 ppm (**d**). Each panel shows the deconvoluted component spectra and, in inset, the measured spectra, averages from two independent batches with similar spectral features; temperature, 5 °C.

**Figure 4 cells-10-02354-f004:**
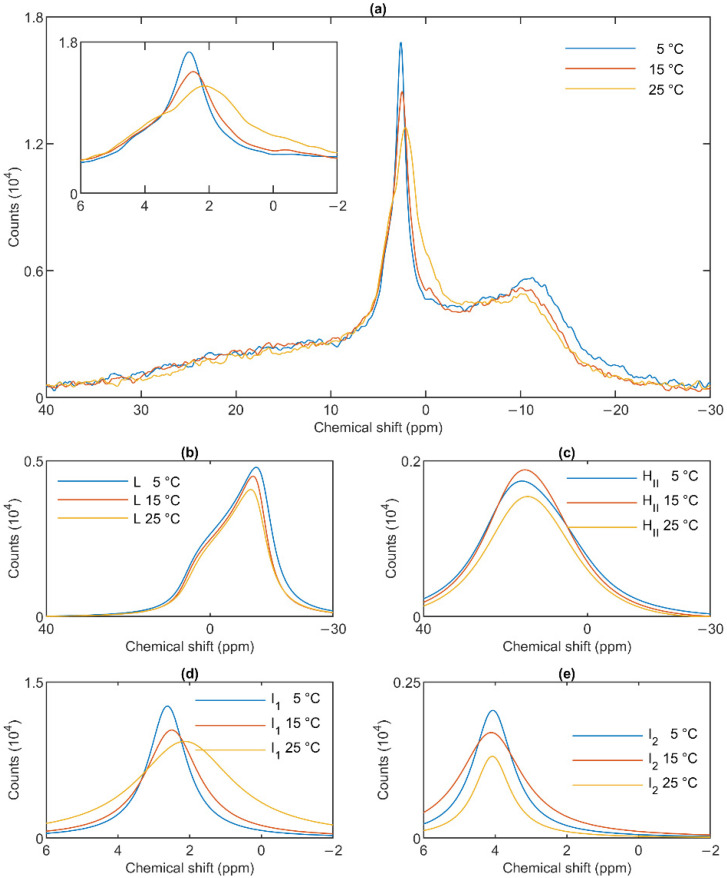
^31^P-NMR spectra of granum thylakoid membranes at different temperatures (**a**) and their component spectra showing variations in L, the lamellar phase (**b**), H_II_, the inverted hexagonal phase (**c**), and the two isotropic phases, I_1_ (**d**) and I_2_ (**e**). The experiments were performed by gradually increasing the temperature from 5 to 35 °C, with data acquisition times between 1 and 2 h. Four experiments on three independent batches, with similar polymorphic features, were averaged to improve the signal to noise ratio. The inset in panel a shows the isotropic region.

**Figure 5 cells-10-02354-f005:**
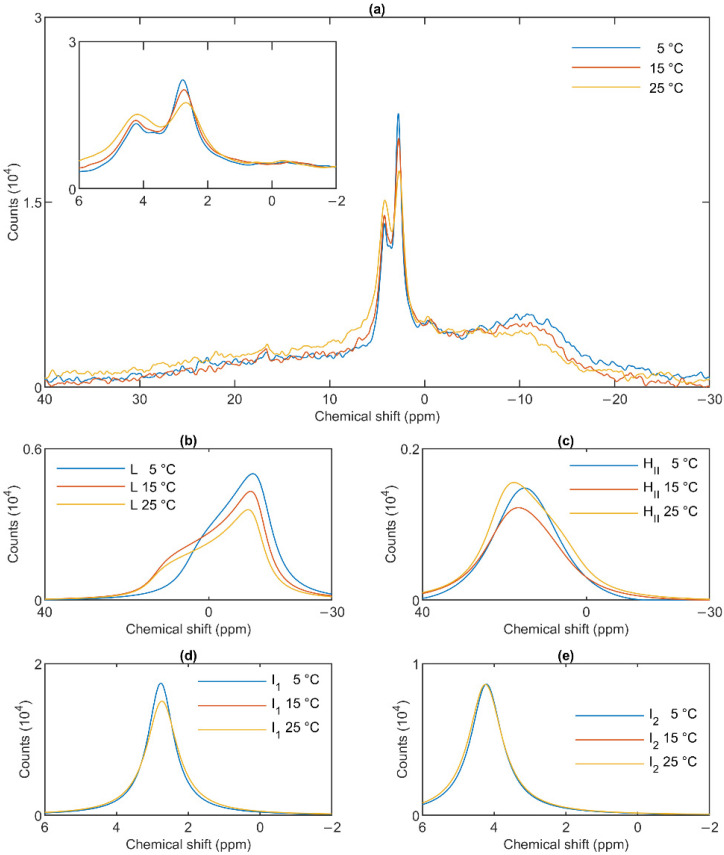
^31^P-NMR spectra of stroma thylakoid membranes at different temperatures (**a**) and their component spectra showing variations in L, the lamellar phase (**b**), H_II_, the inverted hexagonal phase (**c**), and the two isotropic phases, I_1_ (**d**) and I_2_ (**e**). The experiments were performed by gradually increasing the temperature from 5 to 35 °C, with data acquisition times between 1 and 2 h. Two experiments on two independent batches, possessing similar spectra, were averaged. The inset in Panel a shows the isotropic region.

**Figure 6 cells-10-02354-f006:**
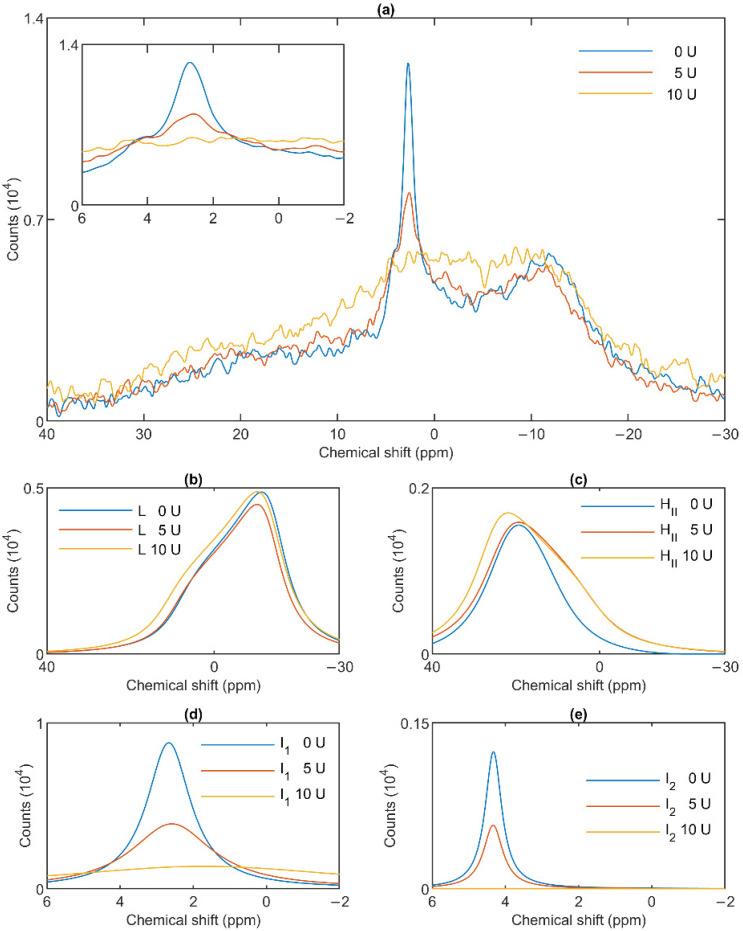
Effects of wheat-germ lipase treatments on the ^31^P-NMR spectra (**a**) of granum thylakoid membranes and on their component spectra: L, the lamellar phase (**b**), H_II_, the inverted hexagonal phase (**c**), and the two isotropic phases, I_1_ (**d**) and I_2_ (**e**). The spectra represent averages of two measurements from two batches exhibiting similar spectra; recorded at 5 °C. The inset in panel a shows the isotropic region.

**Figure 7 cells-10-02354-f007:**
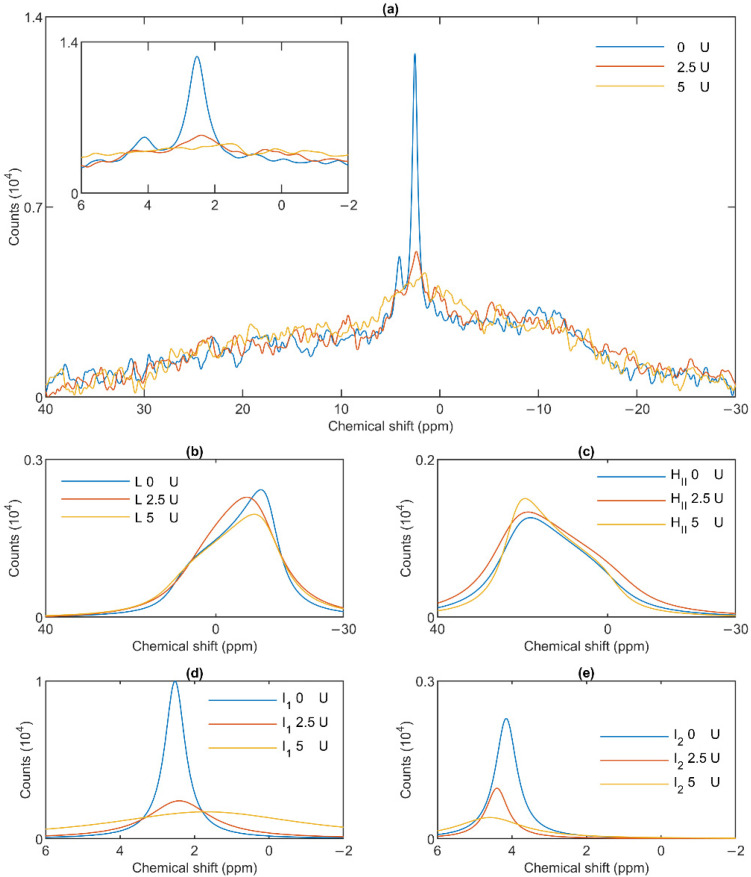
Effects of wheat-germ lipase treatments on the ^31^P-NMR spectra of stroma thylakoid membranes (**a**) and on their component spectra: L, the lamellar phase (**b**), H_II_, the inverted hexagonal phase (**c**), and the two isotropic phases, I_1_ (**d**) and I_2_ (**e**). The spectra were recorded at 5 °C. The inset in panel a shows the isotropic region.

**Table 1 cells-10-02354-t001:** Position and half-bandwidth of the two isotropic peaks of the granum and stroma thylakoid membranes, as derived from the mathematical deconvolution of the ^31^P-NMR spectra recorded at 5 °C. Errors represent standard deviation from 10 experiments of 3 batches.

	Granum		Stroma	
	FWHM	Position (ppm)	FWHM	Position (ppm)
I_1_	1.22 ± 0.16	2.66 ± 0.07	0.71 ± 0.10	2.68 ± 0.16
I_2_	1.51 ± 0.96	3.98 ± 0.40	0.89 ± 0.13	4.19 ± 0.13

## Data Availability

The original data were recorded at the Slovenian NMR Center. Processed and derived data are available from the corresponding author G.G. on request.

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
