# Peer review of "Lipid Polymorphism of the Subchloroplast—Granum and Stroma Thylakoid Membrane—Particles. I. 31P-NMR Spectroscopy"

_cells, 2021, doi:10.3390/cells10092354_

Round 1
Reviewer 1 Report
This paper describes the use of 31P-NMR to establish membranes properties of isolated granum or stroma thylakoid membranes purified from thylakoid membranes. The authors use the same method as they already developed for thylakoid membrane (Dlouhy et al., 2020). They show the presence of four lipid phases: bilayer, hexagonal II and two isotropic phases, as they already found for thylakoid membranes. They didn’t found any obvious difference between their fractions and the intact thylakoid. The authors tried to see if these phases were temperature dependent or lipase sensitive. They did not see any obvious difference of behavior between granum and stroma lamellae. Therefore, they investigate in the paper Part II if some differences could be observed between the two fractions.
This work is based on one major technique P-NMR that has the inconvenient to reflect mainly the distribution of PG that represent only 10 % of thylakoid membranes. Therefore PG could be excluded from some lipid phases with specific rearrangement and some information might be overlooked. Nevertheless, this article is well written and drawbacks are most of the time taken into account.
Major remark:
- The lipase used is a commercial lipase well known to hydrolyze TAG. The specificity of the lipase is not precised in the material and method. Does it work on membrane glycerolipid? Is it working on glycolipids and PG? What is the product of the lipase? The author show that the isotropic phases are sensitive to the lipase whereas the bilayer and hexagonal phases are not. They claim that the isotropic phase are transformed into largely immobile structure. What is the evidence?
- Due to the diversity of spinach harvest and subchloroplast fractionation, the authors explained that the spectral distribution vary from batch to batch. Therefore they choose to represent one spectra that is representative of their results. Therefore it is hard to know if observations are significant. For example, they say line 291-292 “The temperature sensitivities of the different phases are very similar in the granum and stroma TMs” whereas on figure 4 and 5 the behavior of the lamellar contribution seems to be affected differently between the two fractions. At the opposite, line 329-330, it is written “The lipase-susceptibility difference between the I1 and I2 phases was somewhat less marked than in the granum TMs” whereas the I1 phase behavior seems pretty similar in Figure 6 and 7.
Reviewer 2 Report
The MS by Garab and his colleagues present a nice and interesting piece of research about lipid polymorphism and its investigation by solid-state 31P NMR. They have chosen interesting membranes originating from the subchloroplasts, namely granum and stroma thylaloid membranes. Main focus was non non-bilayer forming lipids with focus on hexagonal inverted phases. NMR is ideal to study transition between various lipid phases due to the phase characteristic wideline NMR lineshapes. Using temperature dependent studies they even observed changes between various phases including the temperature-dependent appearance of isotropic phases.
The authors show nicely that those membrane have four different lipid phases and have common polymorphisma despite quite different protein composition. They could even found structurally different identities in those membranes by using an elegant lipase treatment assay. Together with additional data they provide a good and comprehensive description of the structural identities and lipid phase origin and temperature-dependent behavior, all nice information to understand the self-assemblies of those Thylakoid membranes better.
There is one major point which the authors might address in their discussion:
What is the function of TM to have those phases and special structurally identities. There one might perhaps include some examples or speculations about the functional necessity to have those lipid compositions and behavior.
Author Response
We are grateful to Reviewer II for carefully reading our manuscript and for his/her helpful comment, which encouraged us to expand the discussion on the possible roles of non-bilayer lipids and non-lamellar lipid phases in the structure and the structural and functional plasticity of thylakoid membranes. Please find our response below.
Comment: “What is the function of TM to have those phases and special structurally identities. There one might perhaps include some examples or speculations about the functional necessity to have those lipid compositions and behavior.”
Response: We gladly add a paragraph – facts, hypotheses and speculations – to summarize the presently available literature data and our views on these enigmatic questions. This paragraph is appended to the section Conclusions of Part II, which has now a title as Conclusions and Perspectives. We refer to this section at the end of Part I.
We trust that this addition improves our paper and that Reviewer II and the Editor will find our paper suitable for publication in Cells.
Sincerely, Győző Garab on behalf of all co-authors.
Round 2
Reviewer 1 Report
The authors improved their manuscript following the revision however they are still too assertive in their conclusions.
For the first comment, I’m convinced about PG distribution in the thylakoid membrane and even if some information might be missed, this technique enhances different lipid phases that could not be seen otherwise. Furthermore the author’s explanation about spectra averaging is much clearer and the moderation of the significance of their result is well appreciated.
However, for the interpretation of the lipase results, I’m still puzzled. If the lipase is a TAG, DAG or MAG lipase, it might not work on PG or galactolipids. The isotropic phases are affected by the lipase treatment. Could these phases be due to lipid droplet that should be rich in neutral lipid that would be sensitive to the lipase? Furthermore, could the broadening of the isotropic due only to its disappearance and not to the “emergence of a larger formation composed of released or partially cleaved PG” ? The authors did not analyze the effect of the lipase on thylakoid lipids and therefore could not claim the sentence line 341-343. The paragraph on the effect of wheat germ lipase is too assertive and the conclusion should be moderated if the lipase activity on the thylakoid lipid is not further investigated.
Round 3
Reviewer 1 Report
The authors reply to most of my concerns.